# Experimental Analysis of Velocity Distribution in a Coarse-Grained Debris Flow: A Modified Bagnold's Equation

**Donatella Termini [1,*] and Antonio Fichera [2]**

1   Department of Engineering, University of Palermo, 90128 Palermo, Italy
2   Faculty of Engineering, University Enna Kore, 94100 Enna, Italy; antonio.fichera@unikore.it
*   Correspondence: donatella.termini@unipa.it; Tel.: +39-09123896522

**Abstract:** Today, Bagnold's theory is still applied to gravity-driven flows under the assumption of uniform sediment concentration. This study presents findings of flume experiments conducted to investigate the velocity and concentration distributions within the debris body by using high-resolution images. The analysis has shown that the concentration and mobility of grains vary along the depth. A linear law to interpret the grains concentration distribution, starting from the knowledge of the packing concentration and of the surface concentration, Cs, has been identified. By considering such a law, modified expressions of the Bagnold's number and the velocity in stony-type debris flows are also presented. By using these expressions, three regimes of motion have been identified along the depth, and the velocity profile within the debris body is determined as a function of the parameter Cs. It has been verified that the velocity profiles estimated by using the modified equation compare well (mean square error less than 0.1) with the literature's measured profiles when Cs is correctly measured or estimated. Results of cutting tests, conducted for a sample of the used material, have also allowed us to verify that Cs could be determined as a function of the static friction angle of the material.

**Keywords:** debris flows; flow velocity; sediment concentration; prevision

## 1. Introduction

Coarse-grained debris flows represent the gravity-driven motion of granular sediment, which especially occurs in steep valleys and rivers [1,2]. These flows consist of rapid movements of mass that might determine forces of high-impact and, consequently, serious hazards, loss of human life, and structural damage.

The identification of laws governing hydrodynamics of flows in steep river reaches is less advanced than in lowland river systems [3]. This is especially related to the fact that different flow behaviors can be obtained, depending on the concentration and composition of sediments. Consequently, a wide range of coarse-grained debris flows, characterized by different internal stress distributions, can be observed [4–6]. The sediment transport processes in flows characterized by small sediment concentrations have been largely analyzed, and different approaches to treat the bed load and the suspended load can be found in the literature, among others [7–10]. In flows characterized by high sediment concentrations, such as in debris flows, these approaches are generally unsuitable [6]. It should also be considered that, depending on the composition of the mixture determining the dominant effect (i.e., either the inertial stress due to the interstitial fluid or the collision stress due to coarser sediments) in flow dynamics, different regimes of motion can be obtained [11–14]. According to Takahashi's [14] classification, bedload or suspended load and small values of collision stress can

occur when the sediment concentration is less than about 0.02. When sediment concentration ranges from 0.02 to 0.2, the collision stress dominates only in the lower mixing layer of the debris body, and the so-called immature debris flow develops. For values of sediment concentrations larger than 0.2, collision stress becomes important in the entire depth, and the dynamic debris flow develops. In this last case, the grain mobility changes according to the quantity of finer sediment in the mixture. In particular, the stony-type debris flows, characterized by a low portion of finer sediment and a high percentage of coarse sediment (40%–75%), have a large resistance to motion and are dominated by collision stresses that are responsible for the dispersion of grains in the entire flow depth. In muddy-type debris flows and in viscous debris flows, which contain higher percentages of fines and a low portion of coarse sediment (around 10%–30%), the turbulent mixing stresses and the viscous stresses dominate, respectively [14,15].

The first conceptual model for interpreting the dynamics of water/sediment mixtures was formulated by Bagnold [16], who conducted experiments in a concentric cylinder rheometer to examine the effects determined by the dispersion of grains on flow by simultaneously measuring the acting shear and normal forces. Although Bagnold's [16] theory was based on simplified and restrictive experimental conditions (see as an example the exhaustive critical review of [17]), it introduced important and still actual concepts, such as the existence of two different regimes of flow motion (i.e., the macro-viscous and grain-inertia regimes). To identify the two regimes of motion, Bagnold determined a no-dimensional parameter, $N_{Ba}$, that is the ratio between the inertial grain stress and the viscous shear stress. The results of Bagnold's measurements were then used to analyze different processes such as the grains sorting, the flows in gravel beds, and the critical conditions under which a particle remains suspended [18,19]. Takahashi [14] considered a uniform layer of granular loose material and applied Bagnold's [16] constitutive equations to stony-type debris flow by integrating them under the assumption of constant grain concentration in the flow depth.

Especially in recent years, researchers have increased their interest in coarse-grained flows, attempting to identify more convincing conceptual models.

Many theoretical analyses to interpret the dynamics of coarse-grained mixtures can be found in the literature. Different aspects influencing the stress distribution and the motion mechanisms have been investigated, devoting particular attention either to debris flow generated along steep valleys by high run-offs (see, as examples, in [4,15,20–25]), or to the additional bedload transport occurring when runoff-generated debris flows entrain in erodible-bed channels (see, for example, [3,26]), or to the transition processes from the intense bedload transport due to turbulence (as it could occur in coarse beds) to dilute suspension, which also produce the bed and suspended loads in granular flows [27,28]. Some researchers [29–32] have also proposed to apply the kinetic theory to granular flows. Such an approach shows limited applicability because it requires large dimensions of the control volume, compared either with the sediment particle size or with their mean distance and is based on the equation of state, which is not directly related to the shear rate [6,32].

Due to the complexity of the fluid/sediment interactions in debris flow, the elaborated models generally include several empirical parameters and are not easily generalizable.

However, it is clear that the identification of the internal velocity distribution is a key feature in analyzing the dynamics of coarse-grained flow and in estimating the consequent impact forces.

To this aim, several experimental investigations have been conducted, especially in laboratory flumes, by using different typologies of course material, generally artificial, under different hydrodynamic conditions. Lanzoni [33] conducted a series of flume experiments to analyze the velocity profiles of a debris flow formed by a loose sediment layer either of crusher gravels or of glass spheres. Armanini et al. [34] reproduced the granular–liquid mixture by using artificial particles and analyzed the propagation phenomenon of mature debris flow under temporally and spatially uniform conditions. Sanvitale et al. [35] measured velocity profiles of a propagating fluid mixture, consisting of hydrocarbon oil and borosilicate glass of non-uniform size, over a fixed-bed laboratory flume. Iverson et al. [36] conducted experiments in a large-scale flume, releasing a saturated heterogeneous mixture of

sand–gravel and sand–gravel–mud over a rough-fixed bed. Sarno et al. [37] measured the velocity profiles of a steady mixture consisting either of well-sorted round-shaped silica sand or of artificial plastic beads of constant diameter. Hsu et al. [38] and Kaitna et al. [13] considered different gravel mixtures saturated by water and mud. More recently, Fichera et al. [39] analyzed the velocity profiles within a debris flow consisting of crushed gravels with assorted diameters.

Although the aforementioned studies have allowed significant progress in understanding the velocity distribution and the different rheological behaviors of the coarse-grained mixtures, their dynamics have still not been completely defined today. This could be due both to the difficulty of reproducing the real propagation conditions of these flows and to the difficulty in identifying the characteristics of the fluid/sediment mixture accurately [40,41].

In such a context, although Bagnold's theory was based on simplified experimental conditions, inducing high uncertainty in the flow velocity estimation, it is still widely applied to gravity-driven flows, especially in numerical models. The point is that, while Bagnold's results were obtained with fixed concentrations and shear rates [14,16], the studies conducted in this field highlight that the velocity distribution, which depends on the variation of the shear rate within the debris body, can be in turn affected by the variation of the grain concentration in the flow depth.

This suggests that the major problem in the application of Bagnold's theory to gravity-driven flows could be related to the assumption of uniform sediment distribution in the entire depth of flow [42,43].

To the authors' knowledge, no systematic research has been conducted to explore the variation of the grain concentration within the debris body and its effect on the velocity profile. This information could be especially used for interpreting the flow dynamics in numerical models.

The present work is inspired by the aforementioned consideration. In particular, focusing the attention on stony-type debris flows, which are frequent in the Dolomites (Italian Alps) and in alpine catchments, especially during the summer period when high-intensity rainfall storms occur [44–46], the present study aims (1) to gain some insights on the variation of grains concentration within the debris body and its effect on the velocity profile; (2) to explore the applicability of Bagnold's theory to debris flows, according to Takahashi [14,42,43], by removing the hypothesis of uniform sediment concentration. This would allow us to obtain more accurate results in the application of Bagnold's theory to stony-type debris flows.

The analysis is conducted with the aid of data both appositely collected in a laboratory flume and available in the literature. The paper is organized as follows: Section 2 summarizes peculiar aspects of the application of Bagnold's theory to debris flows and describes the experimental apparatus; Section 3 presents the experimental results; discussion is reported in Section 4; finally, conclusions are drawn in Section 5.

## 2. Material and Methods

### 2.1. Pertinent Aspects of Bagnold's Theory and Its Application to Debris Flow

Before proceeding further, it seems important to briefly summarize here some peculiar aspects of Bagnold's theory and its application to debris flows.

As mentioned in the introduction, Bagnold, on the basis of experimental measurements and physical arguments, introduced the concept of two different regimes of motion: the "macro-viscous" regime, which occurs for small shear rates and is dominated by the viscosity, and the "grain-inertia" regime, which occurs for larger shear rates and is dominated by the inertial forces. To distinguish the two aforementioned regimes, Bagnold defined the dimensionless number (i.e., the so-called Bagnold's $N_{Ba}$) given by the ratio between the stresses due to the inertial forces and those due to the viscosity:

$$N_{Ba} = (\rho_s \gamma d_p^2 \lambda^{1/2})/\mu_f \tag{1}$$

where $d_p$ indicates the particle diameter, $\rho_s$ is the density of grains, $\mu_f$ is the fluid's dynamic viscosity, $\gamma$ is the shear rate, and $\lambda$ is the linear concentration of grains given by the following relation:

$$\lambda = \frac{1}{[C_*/C]^{1/3} - 1} \qquad (2)$$

In Equation (2), $C$ is the grain concentration, and $C_*$ is the maximum possible concentration (i.e., the so-called packing concentration), which can be obtained in static conditions where the friction between the sediment particles is of primary importance [47,48].

Bagnold established that the macro-viscous regime occurs for very low Bagnold's numbers (i.e., for $N_{Ba} < 40$), while the grain-inertia regime occurs for high values of the Bagnold number (i.e., for $N_{Ba} > 450$). The intermediate-range of the Bagnold numbers indicates a transitional regime between the aforementioned ones. While in the macro-viscous regime, the stresses are determined by the interaction of the granular phase with the interstitial fluid, in the inertia regime, the inertia associated with the individual grains is more important.

The point is that Bagnold's results were obtained with fixed concentrations and shear rates [6,14,16]. In accordance with Iverson [4], this would imply that the grain concentration should be specified rather than determined by the grain interaction mechanisms.

According to Takahashi [14,42,49], by considering a stony-type debris flow, for which the effect of the interstitial fluid would be negligibly small, and by assuming uniform grain concentration and shear stress proportional to the square of the vertical gradient of the longitudinal velocity $u$, the integration of Bagnold's constitutive equations (under the boundary condition $u = 0$ at $z = 0$, with $u = $ longitudinal velocity and $z = $ direction orthogonal to the bed) gives

$$u = \frac{1}{d_p}\left[\frac{g\cos\theta}{a_i\cos\alpha_i}\right]C\left(1 - \frac{\rho}{\rho_s}\right)^{1/2}\frac{1}{\lambda}\left[h^{3/2} - (h-z)^{3/2}\right] \qquad (3)$$

with

$$C = \frac{\rho\tan\beta'}{(\rho_s - \rho)(\tan\alpha - \tan\beta')} \qquad (4)$$

where $g$ is the gravity acceleration, $\beta'$ is the inclination, $h$ is the local flow depth, $\rho$ is the fluid density, $a_i$ is the so-called "friction coefficient" [16,50], and $\alpha$ is the collision angle. According to Bagnold's experimental tests, it can be assumed $a_i = 0.04$ and $\tan\alpha = 0.75$; Takahashi [14,49] suggested to assume $a_i = 0.42$ for loose beds.

Equation (3) (with $\lambda$ estimated by Equation (2)) depends both on grain concentration $C$ and on the maximum possible concentration $C_*$. While $C_*$ could be estimated by using either experimental data appositely collected or physically-based literature data, more investigations should be conducted to evaluate grain concentration $C$ and its variation within the debris body.

## 2.2. Experimental Apparatus and Procedure

In order to analyze the velocity and grains concentration distributions within the debris body, data collected in a straight reach of a laboratory flume, constructed at the Department of Engineering, University of Palermo (Italy), have been used. The channel was 0.17 m wide and with Perspex transparent side-walls (see Figure 1). For the experimental run considered in the present work, the bed slope was 15° and consisted of assorted gravels with mean diameter $D_{50} = 3$ mm. These experimental conditions were selected after preliminary setting runs and in accordance with the indications of previous literature works [33,39,51,52].

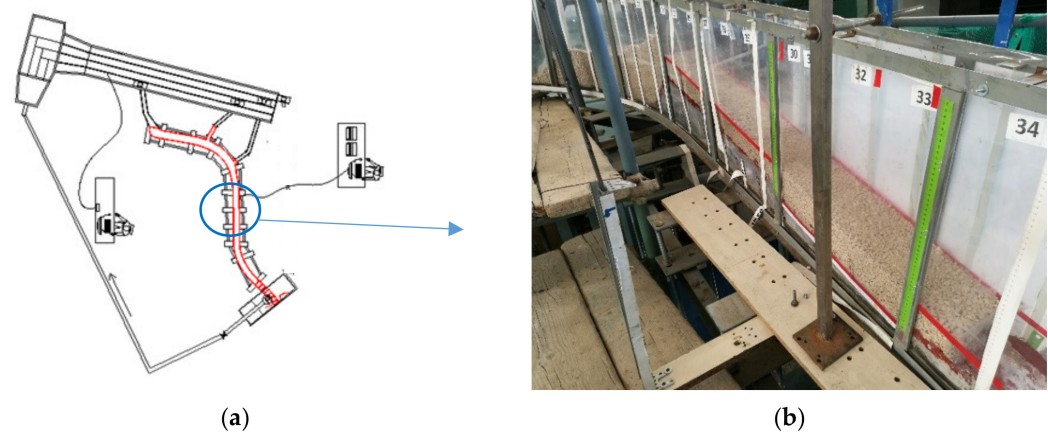

|  |  |
|---|---|
| (**a**) | (**b**) |

**Figure 1.** (**a**) Plan view of the experimental apparatus; (**b**) particular of the channel reach considered for the analysis (bed slope = 15°; assorted gravels with mean diameter $D_{50}$ = 3 mm; the numbers indicate the sections in which the channel was discretized).

Figure 2 reports the grain-size distribution of the bed material, which was determined by using an electric oscillating sieve. The debris flow was generated by releasing the water discharge $Q = 3.5 \times 10^{-3}$ m$^3$/s over a loose sediment layer of a thickness of around 0.10 m, which was previously saturated by slowly releasing a low water discharge $Q_0 = 0.8 \times 10^{-3}$ m$^3$/s. A permeable ground sill was positioned at the downstream end of the examined reach to avoid the degradation of the sediment layer.

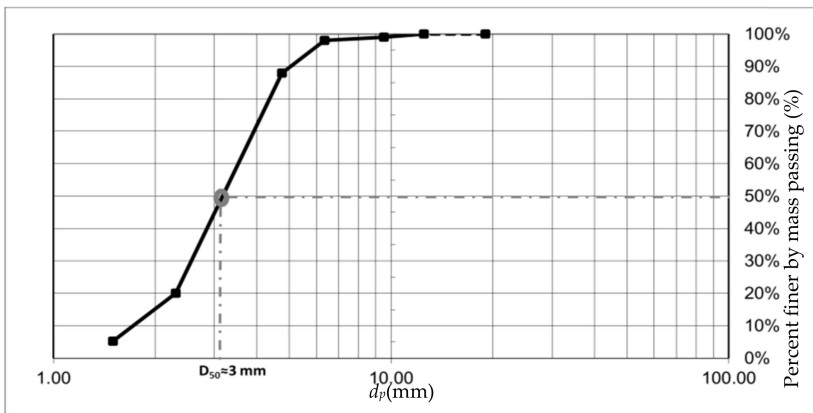

**Figure 2.** Grain size distribution of the bed material (the *x*-axis indicates the grain diameter; the *y*-axis indicates the percent of finer by mass passing).

During the experiment, the water discharge was measured through the ultrasonic instrument Mainstream EH7000 (by Endress+Hauser S.p.a.), which is based on the Doppler effect, located in the upstream inflow channel. A high-resolution digital camera (AOS Technology AG) was used to record the flow motion through the transparent right side-wall. The camera was set at a frame rate of 300 fps, and the investigated area was around 20 cm wide (see Figure 3). The rate of the acquisition frequency was defined, during the preliminary runs, according to the camera's characteristics and the extension of the covered area. During the passage of the debris body, around 600 frames were recorded. Furthermore, three samples of the mixture were taken at the free surface to estimate the free surface grains concentration *Cs*. Each sample was weighed, after having removed the water, with the help of a bake in order to determine the corresponding grain concentration. Finally, the "measured" free surface grain concentration was assumed as the mean value of the grain concentration of the three samples, which was equal to $Cs_{,m}$ = 0.305.

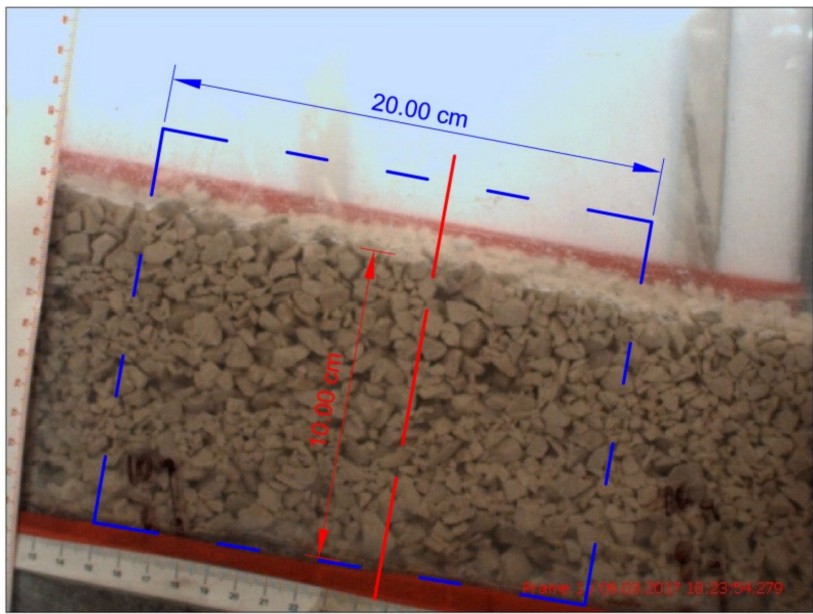

**Figure 3.** Scheme of the investigated area (in blue).

The frames recorded by the high-resolution camera during the passage of the debris body were used to estimate the instantaneous values both of the longitudinal velocity and the grain concentration at different distances, $z$, from the channel bed. To this aim, the sediment layer was divided into sub-layers, parallel to the channel bed, of thickness $St = D_{50}$. Within each sub-layer of thickness $St$, the grain's instantaneous longitudinal velocity, $u(t)$, was estimated as the ratio between the grain's displacement for two consecutive frames (i.e., two consecutive times) and the time interval. In particular, by considering the orthogonal local reference system with an origin at the southwest corner of the frame, it was estimated $u(t) = (s_{i+1} - s_i)/(t_{i+1} - t_i)$ (where $t_{i+1}$ and $t_i$ are two consecutive times and $s_{i+1}$ and $s_i$ are the corresponding grain's positions). Then, the time series of $u(t)$, obtained for the grains within each sub-layer $St$ by considering all the recorded frames, were used to determine both the corresponding time-averaged value $u$ and the mean, $\overline{u}$, of the time-averaged velocities of the considered sub-layer $St$.

The recorded frames were also used to estimate the instantaneous values of the grain concentration within each sub-layer of thickness $St$. To this aim, a four-step procedure, appositely implemented in a Matlab environment and allowing us to count the grains in the considered sub-layer, was applied to each frame. Then, the time series of the grain concentration obtained for each sub-layer $St$ were used to determine the corresponding time-averaged value C. Thus, the grain concentration distribution along the depth was analyzed by using the values of C obtained for all the sub-layers. The sensitivity analysis of the estimated grain concentration distribution with the value of the thickness $St$ was also performed, as explained in the following Section 3.1.

## 3. Results

### 3.1. Grains Concentration Distribution

Both by the direct analysis during the experimental run and as a result of the aforementioned four-steps procedure, it was verified that the grain concentration varies along the direction orthogonal to the bed. In particular, as Figure 4 clearly shows, as one passes from the free surface to the bed, the movement of the grains decreases and the grain concentration increases. Close to the bed, the frictional stress is so high that no motion is possible (static configuration), and the grain concentration assumes its maximum value (i.e., the maximum packing concentration) $C_*$. The thickness of the static bed, $z_0$, was estimated from the analysis of both of the recorded frames and of the measured velocity profiles.

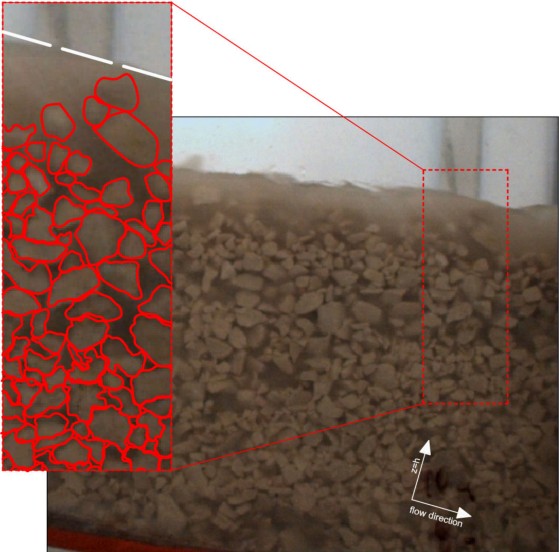

**Figure 4.** Variation of grains concentration in the depth of flow and particular of the grain configuration.

The values of grain concentration $C$ obtained for all the sub-layers were plotted against the distance from the upper level of the static bed, as reported in Figure 5.

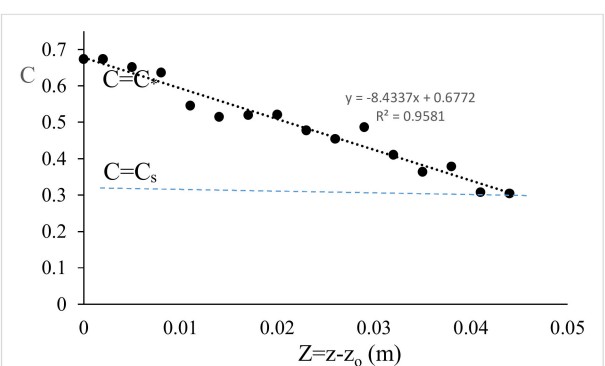

**Figure 5.** Measured grain concentrations $C$ and interpolating law ($Z$ indicates the direction orthogonal to the bed with an origin at the upper level of the static bed).

As Figure 5 shows, the points can be interpolated by a linear law (with regression coefficient $R^2 = 0.96$). The values of the grain concentration at the extreme points of the interpolating line are very close, respectively, to those of the surface concentration, $Cs$, and of the maximum possible concentration $C_*$ at the bed. Based on this, the following linear law is considered to approximate the distribution of the grain concentration

$$\frac{C(z) - C_*}{C_S - C_*} = \frac{z - z_o}{h - z_o} \tag{5}$$

where $C(z)$ indicates the value of the grain concentration at level $z$. Based on Equation (5), the variation of the grain concentration within the debris body can be estimated as a function of the two parameters $C_*$ and $Cs$.

For the present application, it is $C_* = 0.676$ and $Cs = 0.305$, and thus Equation (5) can be written as $C(z) = -8.793Z + 0.68$.

In order to investigate the sensitivity of the estimated grains concentration distribution to the thickness of the sub-layer, other two values of $St$ have been considered (i.e., $St = 2D_{50} = 6$ mm; $St = 3D_{50} = 9$ mm) and the corresponding values of grain concentration $C$ have also been estimated by applying the aforementioned four-step procedure. Then, the values of the grain concentration determined for

each value of *St* were compared with the theoretical distribution given by Equation (5), as reported in Figure 6.

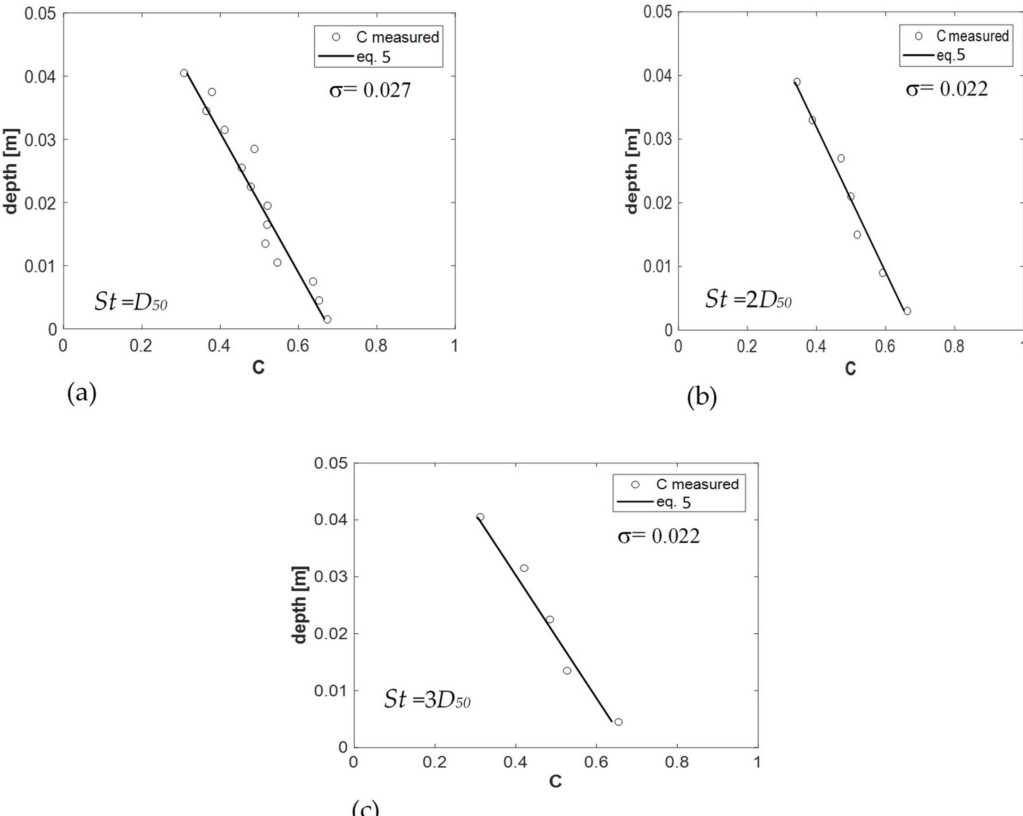

**Figure 6.** Comparison between the theoretical distribution of the grain concentration (Equation (5)) and the estimated values (C measured) by assuming different sub-layer thickness *St*: (**a**) for $St = D_{50}$; (**b**) for $St = 2D_{50}$; (**c**) for $St = 3D_{50}$. The *x*-axis indicates the grain concentration; the *y*-axis indicates the distance from the upper level of the static bed.

Figure 6 also reports the values of the root mean square error ($\sigma$) between the values of the grain concentration calculated in the *i*-th sub-layer by using Equation (5), $C_{i,c}$, and those measured, $C_{i,m}$:

$$\sigma = \sqrt{\frac{\sum_{i=1,N_{ts}}(C_{i,m} - C_{i,c})}{N_{ts}}} \tag{6}$$

where $N_{ts}$ is the number of the sub-layers of thickness *St*. It can be observed that $\sigma$ assumes small and almost equal values for all the considered values of *St*. This suggests that the estimated grains concentration distribution does not depend on the selected value of *St*. Finally, a value of $St = 2D_{50}$ (6 mm) has been assumed for the subsequent analysis.

### 3.2. Modified Bagnold's Equation Applied to Debris Flow

Based on the results obtained in Section 3.1, Equation (3) is rewritten as follows:

$$u = \frac{1}{d_p}\left[\frac{g\cos\theta}{a_i\cos\alpha_i}\right]C(z)\left(1 - \frac{\rho}{\rho_s}\right)^{1/2}\frac{1}{\lambda(z)}\left[h^{3/2} - (h-z)^{3/2}\right] \tag{7}$$

with

$$\lambda(z) = \frac{1}{[C(z)/C_*]^{1/3} - 1} \tag{8}$$

where *C(z)* is given by Equation (5).

In Figure 7, the velocity profile estimated by applying Equation (7) is compared to that determined by applying Equation (3). This figure also reports both the profile of the mean measured velocity $\bar{u}$ (indicated as "experimental profile") and the velocity *u* of selected grains. From Figure 7, it is clear that while the velocity profile estimated by applying Equation (3) (i.e., with uniform grain concentration within the debris body) deviates from the experimental one, the velocity profile estimated by using Equation (7) (i.e., by considering the linear variation of the grain concentration within the debris body) well approximates the experimental profile.

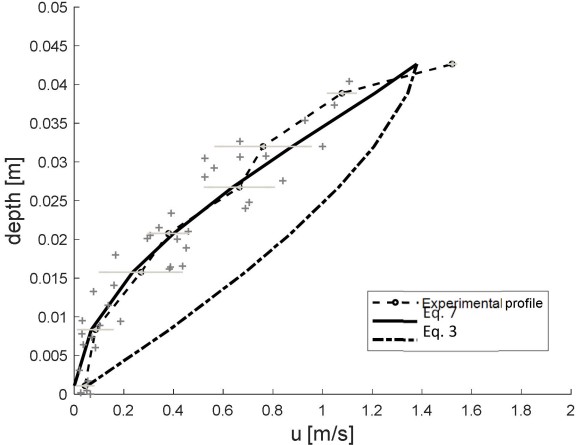

**Figure 7.** Comparison between the $\bar{u}$ profile (experimental profile) and the estimated ones by applying Equation (3) and by applying Equation (7). The dots represent the velocity *u* of selected grains (the *x*-axis indicates the longitudinal velocity; the *y*-axis indicates the distance from the upper level of the static bed).

This result indicates that Equation (5) well interprets the variation of the grain concentration within the debris body, also confirming that the assumption of uniform grain concentration in the entire flow depth is not realistic.

Thus, the Bagnold number $N_{Ba}$ has been rewritten as follows:

$$N_{Ba}(z) = [\rho_s \gamma d_p^2 \lambda(z)^{1/2}]/\mu_f \tag{9}$$

By applying Equation (9), the regime of flow motion has been identified in the entire depth of flow, as reported in Figure 8.

In particular, Figure 8 reports both the $N_{Ba}$-values obtained by using Equation (9), with the velocity and the grain concentration distributions respectively given by Equations (7) and (5), and the $N_{Ba}$-values (hereon indicated as "experimental $N_{Ba}$-values") obtained by using the experimental velocity profile. Figure 8 also reports the $N_{Ba}$-values obtained by using Equations (1) and (3). Figure 8 shows that while the distribution of the $N_{Ba}$-values estimated by using Equation (9) is in agreement with the profile of the experimental $N_{Ba}$-values, the distribution of the $N_{Ba}$-values estimated by using Equation (1) deviates from it. In particular, the last one has $N_{Ba}$-values greater than 1000, close to the bed, and a decreasing trend as one passes from the bed to the free surface. On the contrary, Equation (9) determines $N_{Ba}$-values less than 1000 close to the bed and greater than 1000 close to the free surface. The latter trend of the $N_{Ba}$-values is consistent with that observed in previous literature works [34,48,51], indicating the development of the frictional flow regime for $N_{Ba}$-values <1000 and of the collisional-inertial flow regime for $N_{Ba}$-values >1000.

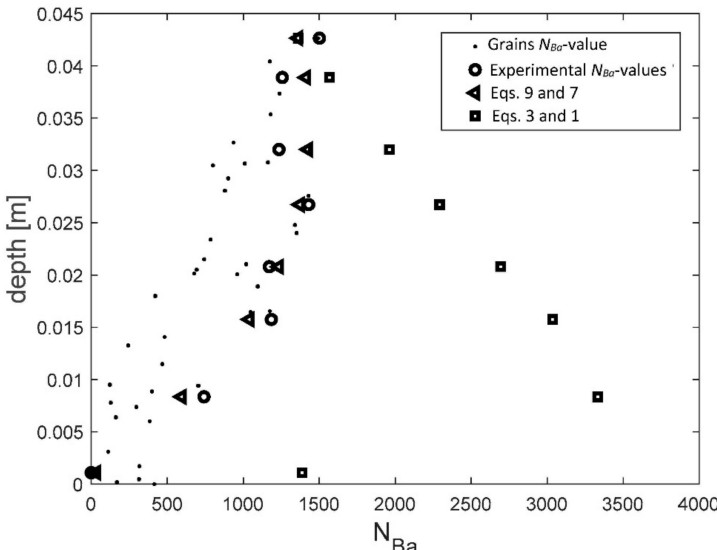

**Figure 8.** Comparison between the $N_{Ba}$-values by using the experimental velocity profile (experimental $N_{Ba}$-values), the $N_{Ba}$-values estimated by applying Equations (9) and (7) and the $N_{Ba}$-values estimated by applying Equations (3) and (1). The dots represent the $N_{Ba}$-values of selected grains; (the *x*-axis indicates the $N_{Ba}$-values; the *y*-axis indicates the distance from the upper level of the static bed).

Based on the obtained results, it can be concluded that, in the present application, three different regimes of motion can be identified within the debris body (see Figure 9): the collisional regime close to the free surface with $N_{Ba}$-values >1000 and $C \cong Cs$, the frictional regime in the intermediate zone with $N_{Ba}$-values <1000 and $C = C(z)$, and the static zone close to the bed with $N_{Ba}$-values <<1000 and $C \cong C*$.

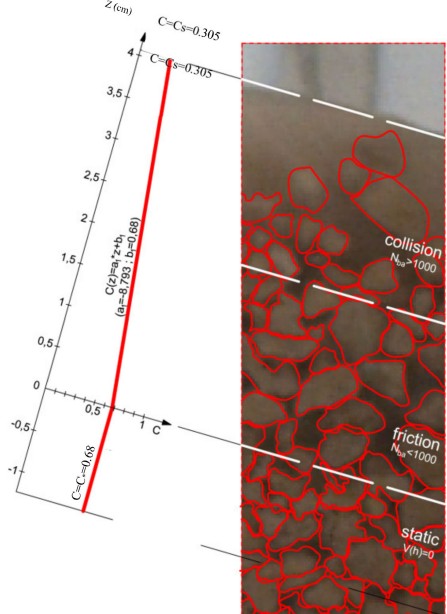

**Figure 9.** Regimes of motion within the debris body.

### 3.3. Free Surface Grains Concentration and Procedure for Its Estimation

Based on Equation (7), the longitudinal velocity profile should be determined as a function of the variation of the grain concentration defined by Equation (5) and, thus, as a function of the parameters $C*$ and $Cs$. While the parameter $C*$ could be easily identified, as mentioned in Section 2.1, it is difficult to determine the free surface grain concentration, $Cs$.

Thus, in this part of the work, experimental velocity profiles taken from the literature, i.e., data from [33,35,37,39] were considered and the parameter $Cs$ was determined by minimizing the mean square error between the velocity values determined by using Equation (7) and the experimental ones:

$$\min[\sigma_v] = \min \sqrt{\frac{\sum_{i=1,n}(u_{m,i} - u_{c,i})^2}{n-1}} \tag{10}$$

where $n$ is the number of the measurement points, $u_{m,i}$ and $u_{c,i}$ indicate, respectively, the measured and the calculated velocity values in the $i$-th sub-layer.

Figure 10 shows the comparison between the values of the parameter $Cs$ obtained as a result of the minimization process (indicated as "optimal" values of $Cs$) and those taken from the literature [33, 35,37,39]. The line of the perfect agreement is also reported in Figure 10. From this figure, it can be observed that, generally, the "optimal" values of $Cs$ are slightly greater than those measured, and a very good agreement between the "optimal" values of $Cs$ and those measured by Fichera et al. [39] can be observed. The different behavior observed in the latter case could be due to the fact that while Fichera et al. [39] measured the free surface concentration during the passage of the debris body in the examined reach, as well as in the present work, the other values of $Cs$ were measured either not during the passage of the debris body or in sections different from those examined or not at the free surface. This further indicates the importance of the correct definition of the parameter $Cs$.

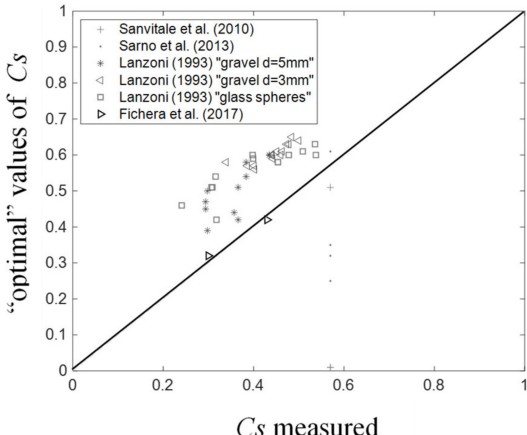

**Figure 10.** Comparison between the estimated "optimal" values of the parameter $Cs$ and those measured and taken from the literature.

In order to identify a more generalizable procedure to estimate $Cs$, it has been taken into account that, according to Armanini et al. [34] (see also in [51]), in a flowing layer of grains over a loose bed of inclination β, fully saturated with water, and, with grain concentration assumed coincident with the transport concentration $Cs$, it is

$$\tan\beta = \left[\frac{(\rho_s - \rho)C_s}{\rho + (\rho_s - \rho)C_s}\right]\tan\varphi_c \tag{11}$$

where $\varphi_c$ represents the critical friction angle.

For negligible interstitial overpressures (as it occurs at the free surface), in Equation (11), $\varphi_c$ could be assumed equal to the static friction angle of the material. This means that Equation (11) would allow us to obtain information on the grain concentration $Cs$, starting from the knowledge of the bed slope and of the static friction angle, which can be determined by simple shear tests.

Thus, the existence of the relationship (Equation (11)) between the static friction angle and the concentration $Cs$ is explored for the present application. To this aim, the static friction angle $\varphi_c$ was

determined by using the direct cutting apparatus available at the Geotechnical Laboratory of the Engineering Department, University of Palermo. Such an apparatus is specially adapted to the present application because it is equipped by a large cutting box ($30 \times 30 \times 20$ cm$^3$—see Figure 11), allowing us to test a sample, of adequate volume, of the material used for the experimental run.

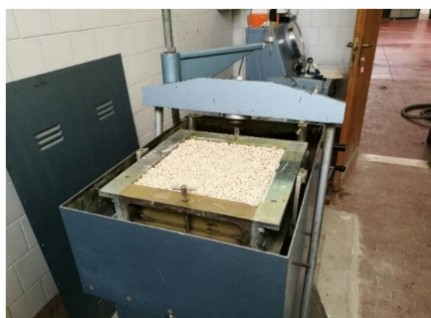

**Figure 11.** Box used for the cutting tests.

The cutting tests were performed, at a constant volume and in saturated conditions, for three samples of the debris body by inducing a constant horizontal force. As a result, the relation between the cutting shear stress, $\tau$, and the corresponding cutting force, $\sigma_n$, was obtained for each test. More details of the testing procedure can be found in Fichera [53]. From the cutting tests, it was also verified that all the examined samples assumed a very similar behavior. By considering the maximum value of the shear stress, $\tau_{max}$, of each test, the couples of values [$\tau_{max}$, $\sigma_n$] were reported on a Cartesian plane. As Figure 12 shows, the coupled values [$\tau_{max}$, $\sigma_n$] are arranged around an interpolating line, having an angular coefficient equal to $\varphi_{c,e} = 38°$. Such an angular coefficient represents the "experimental" value of the static friction angle of the used granular material.

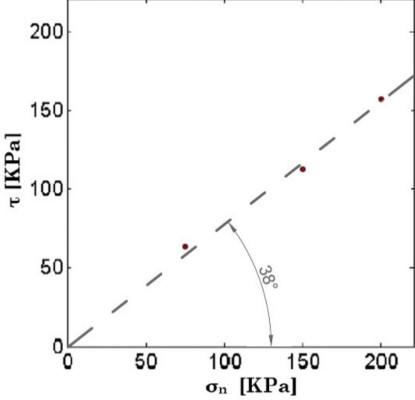

**Figure 12.** Coupled values [$\tau_{max}$, $\sigma_n$] and interpolating line.

Then, the static friction angle was also determined from Equation (11) by considering that, for the examined case, the measured surface grain concentration is $Cs_{,m} = 0.305$ and the inclination is $\beta = 15°$. Finally, a value $\varphi_c = 38°$ was obtained from Equation (11). It is clear that this value of $\varphi_c$ is equal to that obtained from the aforementioned direct cutting tests.

Thus, it can be concluded that, in the first approximation, the free surface grain concentration, $Cs$, can be estimated by applying Equation (11), starting from the knowledge of the bed slope and of the static friction angle of the material.

## 4. Discussion

*4.1. Velocity Profiles and Comparisons with Literature Data*

The applicability of the Equations (5) and (7) was also assessed by comparing the estimated velocity profiles with measured velocity profiles taken from the literature. In particular, measured profiles taken from Lanzoni [33] (see also in [53,54]), Sanvitale et al. [35], Sarno et al. [37], and Fichera et al. [39] were considered. In total, measured profiles from 46 pieces of literature were used for the comparison.

In Figure 13, the velocity profiles estimated by using Equations (5) and (7) are compared with those measured. In particular, this figure reports the velocity profiles estimated by assuming for the parameter *Cs* both the values taken from the literature and the calculated "optimal" values.

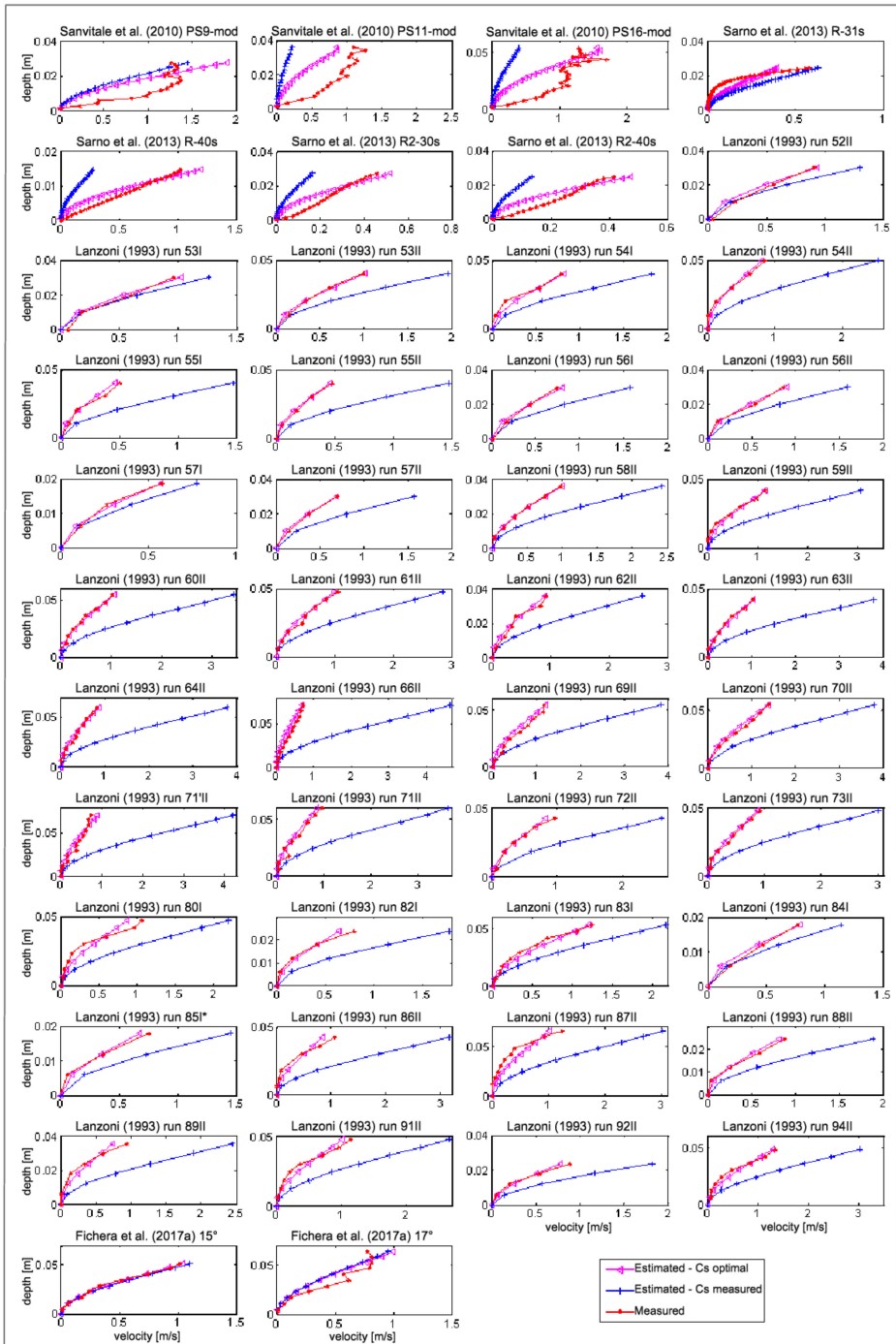

**Figure 13.** Comparison between the estimated velocity profiles and those taken from the literature.

It can be seen that, while in the first case the estimated velocity profiles differ from those measured when the "optimal" values of *Cs* are used, the estimated velocity profiles compare well with the measured ones and, with the exception of the comparisons with SanVitale et al.'s [35] data, a value of the mean square error less than 0.1 is obtained. The differences obtained within the debris body between the estimated velocity profiles and those measured by SanVitale et al. [35] could be related to the fact that SanVitale et al. [35] analyzed a fluid/sediment mixture (consisting of borosilicate glass and hydrocarbon oil) of chemical/physical characteristics different from those examined by the other authors. This confirms previous literature findings indicating that the chemical/physical characteristics

of the material could affect the stress distribution and, thus, the behavior of the fluid/sediment mixture (see, as examples, [55,56]).

Thus, on the one hand, Figure 13 demonstrates that Equation (7) allows us to correctly interpret the velocity distribution within the debris body by taking into account the grain concentration variation through Equation (5); on the other hand, it confirms the importance of the correct definition of the parameter *Cs*. Based on the results presented in Section 3.3, *Cs* could be determined either by direct measurements during the passage of the debris body or by applying Equation (11) as a function of the static friction angle of the material.

### 4.2. Main Aspects Derived by the Proposed Modified Expressions and Procedure

As mentioned in the introduction, in natural environments, stony-type debris flows can be generated in rapid valleys and in high slope streams [1,2]. In these types of debris flows, in which finer sediments do not affect the overall behavior of the mixture, the assumption of the grain concentration being uniform everywhere is generally accepted in analyzing flow dynamics [11,12,14].

The analysis conducted in the present work has substantially highlighted that the identification of the grain concentration distribution within the debris body as an important aspect in estimating the velocity profile. In particular, the presented experiment has provided useful information on the distribution of the grain concentration within the debris body. It has been observed that as one passes from the free surface towards the bed, the grain concentration increases, and the movement of the grains decreases. At the bed, the frictional stress is so high that no motion is possible, the static configuration is obtained, and the grain concentration assumes its maximum value (i.e., the so-called maximum packing concentration), $C_*$. This confirms that the assumption of uniform sediment distribution in the entire depth, which is generally used in analyzing the hydrodynamics of stony-debris flow, is not realistic.

The analysis of the measured values of the grain concentration has shown that the grain concentration distribution within the debris body can be approximated by a linear law, which can be identified starting from the knowledge of the maximum packing concentration at the bed, $C_*$, and the value of the grain concentration at the free surface, *Cs*.

Furthermore, the obtained results have shown that the variation of grain concentration along the depth strongly affects the grains' mobility, thus confirming previous literature findings [13,51] suggesting the development of a spatially variable behavior and stress regime within the debris flow.

By considering the obtained linear law of the grain concentration distribution, modified expressions of Bagnold's number $N_{Ba}$ and of the velocity for stony-type debris flows have been obtained.

The analysis of the values of Bagnold's number ($N_{Ba}$), estimated by using the aforementioned modified expression, has highlighted that three different flow regimes can be identified within the debris body. In particular, the collisional regime is obtained close to the free surface, where the grain concentration is $C = Cs$ and the $N_{Ba}$-values are greater than 1000; the frictional regime is obtained in the intermediate zone, where $C = C(z)$ and the $N_{Ba}$-values are lower than 1000; close to the bed, where the concentration is $C = C_*$ and the $N_{Ba}$-values are strongly lower than 1000, the static zone is established. The observed behavior is consistent with results obtained in other literature works (see in [34,51,53]), indicating the development of the frictional flow regime for $N_{Ba}$-values <1000 and of the collisional-inertial flow regime for $N_{Ba}$-values >1000.

The proposed modified expression to estimate the velocity in stony-type debris flows allows us to evaluate the flow velocity profile within the debris body as a function of two parameters, that are the maximum packing concentration, $C_*$, and free surface concentration, *Cs*. While parameter $C_*$ can be easily identified by using physically-based literature data, it is difficult to determine parameter *Cs*. Thus, first, the "optimal" values of parameter *Cs* were calculated by minimizing the mean square error between the velocity values estimated by using the proposed expression and the experimental ones taken from the literature. As is clear from the previous subsection, this analysis has allowed us to highlight the important role of parameter *Cs* in defining the velocity distribution within the debris

body. The comparison between the velocity profiles estimated by considering the "optimal" values of $Cs$ and the literature's measured profiles (from [33,35,37,39]) has demonstrated that the proposed modified expression, which takes into account the variation of the grain concentration within the debris body, correctly interprets the measured velocities.

A useful general procedure for estimating free surface concentration Cs is also provided by using the results of cutting tests, also performed in the ambit of the present study, of a sample of the granular material used for the experimental run. Based on the relationship between the grain concentration and the bed slope, according to Armanini et al. [34], it has been verified that, in first approximation, surface grain concentration $Cs$ could be determined, starting from the knowledge of the bed slope and of the static friction angle of the material, which could be easily estimated by simple direct cutting tests.

In summary, based on the aforementioned considerations, the use of the proposed expressions allows us to easily estimate the velocity profile and the stress regime variation within the debris body according to the procedure shown in the flow chart of Figure 14.

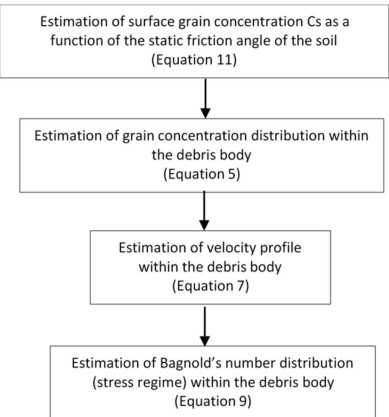

**Figure 14.** Flow chart of the procedure by using the proposed modified equations.

## 5. Conclusions

The present work deals with the dynamics of stony-type debris flows, focusing attention on the velocity and grain concentration distributions. The results obtained can be summarized as follows:

(1) the distribution of the grain concentration can be interpreted by a linear law obtained between the value of the maximum package value, $C_*$, at the bed and the value of the free surface concentration, $Cs$;

(2) by removing the hypothesis of uniform grain concentration along the entire depth, modified expressions of Bagnold's number and of the longitudinal velocity, which take into account the variation of the grain concentration in the entire depth, are presented. The expression of the velocity profile includes two parameters: the maximum package value, $C_*$, which could be determined by using either experimental data appositely collected or physically-based literature data, and the value of the free surface concentration, $Cs$;

(3) by using the modified expression of Bagnold's number, it has been verified that a varying stress regime can develop within the debris flow. The $N_{Ba}$-values are strongly lower than 1000 when close to the bed (frictional regime) and are greater than 1000 (collisional-inertial regime) when close to the free surface;

(4) it has been verified that, in the first approximation, surface concentration $Cs$ can be estimated as a function of the static friction angle of the material, which can be determined by simple shear tests.

In summary, based on the results presented in this work, the velocity and the grain concentration distributions within the debris body can be estimated, starting from the knowledge of parameters $C_*$ and $Cs$ that can be easily identified through physically-based literature data and by simple shear tests,

respectively. This result is of great importance, especially in numerical modeling of stony-type debris flows, which in nature especially occur in rapid valleys or in high-slope streams.

**Author Contributions:** Conceptualization, D.T. and A.F.; methodology, D.T. and A.F.; validation, D.T. and A.F.; formal analysis, D.T.; investigation, A.F.; resources, D.T. and A.F.; data curation, A.F.; writing—original draft preparation, D.T.; writing—review and editing, D.T.; visualization, A.F.; supervision, D.T. and A.F. All authors have read and agreed to the published version of the manuscript.

**Funding:** This research received no external funding.

**Acknowledgments:** This work was partially supported by Italian National Research Programme PRIN 2017, with the project "IntEractions between hydrodyNamics flows and bioTic communities in fluvial Ecosystems: advancement in dischaRge monitoring and understanding of Processes Relevant for ecosystem sustaInability by the development of novel technologieS with fIeld observatioNs and laboratory testinG (ENTERPRISING)".

**Conflicts of Interest:** The authors declare no conflict of interest.

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
