# Peer review of "Experimental Analysis of Velocity Distribution in a Coarse-Grained Debris Flow: A Modified Bagnold’s Equation"

_water, doi:10.3390/w12051415_

Round 1

Reviewer 1 Report

Experimental analysis of velocity distribution in a coarse-grained debris flow: a modified Bagnold’s equation

This study has major flaws in clarification of rationale and objectives. It also struggles with intermixing of results and methods. Another huge flaw in this study is the complexity of language and discontinuity of ideas between paragraphs which makes it nearly impossible for readers to read the manuscript thoroughly. Below I have tried to put my major concerns systematically which the authors need to address before I can recommend publication of the manuscript.

Abstract: the abstract lacks synopsis of rationale, objectives and result statistics.

  1. Introduction

The introduction starts with concept of debris flow but why the authors completely have ignored the idea of suspended load and bed load in debris flow. There are several studies which confirm about their interrelationship for eg.

1) Egashira and Ashida (1992) “Unified view of the mechanics of debris flow and bed-load” Studies in Applied Mechanics.

2) Theule, J. I., et al. "Channel scour and fill by debris flows and bedload transport." Geomorphology 243 (2015): 92-105, etc.

Further some studies related to bedload and suspended load.

1) Van Rijn, Leo C. "Sediment transport, part I: bed load transport." Journal of hydraulic engineering 110.10 (1984): 1431-1456.

2) Joshi, S., & Xu, Y. J. (2017). Bedload and suspended load transport in the 140-km reach downstream of the Mississippi River avulsion to the Atchafalaya River. Water9(9), 716.

3) "Sediment dynamics in the lowermost Mississippi River." Engineering geology 45.1-4 (1996): 457-479.

4) Joshi, S., & Jun, X. Y. (2018). Recent changes in channel morphology of a highly engineered alluvial river–the Lower Mississippi River. Physical Geography39(2), 140-165].

5) Nittrouer et al. (2012) Mitigating land loss in coastal Louisiana by controlled diversion of Mississippi River sand (nature geoscience).

6) Nittrouer and Viparelli 2014 Sand as a stable and sustainable resource for nourishing the Mississippi River Delta (nature geoscience).

7) Allison, Mead A., et al. "A water and sediment budget for the lower Mississippi–Atchafalaya River in flood years 2008–2010: implications for sediment discharge to the oceans and coastal restoration in Louisiana." Journal of Hydrology 432 (2012).

8) Joshi and Xu (2015) Assessment of suspended sand availability under different flow conditions of the Lowermost Mississippi River at Tarbert Landing during 1973-2013 (Water).

I recommend the authors to reference these and other required papers to make a necessary connection between debris flow and sediment transport procedure in rivers and then narrow down to debris flow. This will also help emphasize on the specific use of Bagnold’s equation in this study.

Furthermore, the whole introduction is quite unsystematic, random, and completely lacking interconnections between paragraphs. The rationale of the study is not convincing at all. Also, as a reader I also had a hard time in grasping the connection between title, objectives, and introduction. They have mentioned nothing about sediment concentration earlier which is one of their sub-objective. What type of insights you want to gain (be specific)? Furthermore, it has not been explained about why the authors are focusing on stony-type debris flow? What exactly does stony type debris flow means?

Also, in objective 2 the authors mention about exploring applicability of Bagnold’s theory by removing the hypothesis of uniform sediment concentration? But nowhere before this hypothesis has been explained. Please make introductory comments by briefly explaining the Bagnold’s theory beforehand

Regarding the rationale, the authors have compiled a list of studies linked to debris flows in one way or other but they fail to connect their idea of enlisting those studies to the objectives of this study. I guess they want to state that velocity distribution analysis in debris flow has not been carried out yet, however, they completely fail to give this take home message. So, can you please restructure the whole introduction accordingly? In addition, I am also not convinced by simply saying lack of studies was the reason behind this study? So what? What meaningful support this study can give to the mechanism of debris flow in rivers? How will the modification help Bagnold’s equation in the bigger picture of sediment transport?

  1. Material and Methods

P3L113 what exactly is stony-type debris flow? Please explain its statistical parameters with references.

P3L113 what is the basis for this assumption?

P3L115 what does ‘z’ mean? Why are the boundary conditions set to be ‘0’ for both ‘u’ and ‘z’?

P4L130-132 It looks like the authors are trying to investigate for ‘maximum possible concentration’ but have they clarified this in the objectives? Also, how will the analysis on maximum possible concentration justify the distribution of sediment concentration? Also, how does the velocity distribution fit in here?

What is the basis for/reason behind using the specific values for parameters given in section 2.2? For e.g. Bed slope of 15 degree, diameter 3mm, Discharge 3.51/s (3.51?? – is it cubic m?), sediment layer thickness of 0.1, etc.

It will be better if the authors can get a wider frame for Figure 1 including most of the experimental flume (as much as possible at least). Also please describe the settings in the captions including the meaning for the numbers 31, 32, 34, etc.

P4L150 please explain EH700 and a brief working mechanism behind it.

P4L153 how did you define the width of investigating area?

In figure 2, please elaborate the caption by explaining the x and y-axes. Also please enlarge figure 2 such that numbers and axes titles are seen more clearly. Also I am not sure if the mechanism behind Figure 2 has been clearly explained anywhere.

After a complete reading up to page 5 of the manuscript I have a slight idea of what the authors are trying to do. However, can they at least restructure the ideas such that the message is clear in the introduction leading up to the objectives and methods should only be a reflection of their experiment? I believe section 2.1 must be restructured by specifying what is meant to do at first.

Also, the methods for investigating velocity distribution are either missing or hidden somewhere in section 2.2, can you highlight them clearly? This is one of the main objectives of your study.

Also, the methods say nothing about what modifications will be carried out in the Bagnold’s equation although this is mentioned even in the title.

  1. Results

P6L181-186 these are more assumptions than actual quantifications. Also what do you mean by static configuration?

P6L187-194 these are actually methods according to me, not results

Where is equation (5)?

To me section 3.1 mostly presents methods rather than results. I would like to see the actual quantification of grains concentration distribution here rather than the theory and equations behind the distributions.

Results section for showing the statistics which have been revealed after analysis and not the equations. For e.g. a detailed explanation of Figure 5 will be results but the mechanism behind it will certainly highlight methods.

Please elaborate all the notations in figure 5 in the caption (please do this for all figures consistently).

The same problem of intermixing between results and methods persists throughout the rest of the results section to me. Further problem also exists on lack of or small systematic description on the figures which highlight most of the results.

Also, a discussion section needs to be separated from conclusions for discussing about implications and importance of results that you report in your study, comparing your results with other study results, what they are missing, how your study fills the gap, also in your study what is the importance of the results, what is the reason behind getting those results.

Author Response

Reply to reviewers

First of all, Authors wish to thank reviewers for their comments and suggestions which allowed the improvement of the manuscript in its revised form

Original comments by the reviewers are in normal font; reply is in blue and italics. The numbers indicating the references cited in this reply (and not added here) can be found in the revised manuscript 

Reviewer 1:

Reviewer

Abstract: the abstract lacks synopsis of rationale, objectives and result statistics.

Reply:

Abstract has been improved and partially re-written in order to better highlight the objectives, results and results statistics in terms of mean square error, as suggested.

Reviewer

Introduction

The introduction starts with concept of debris flow but why the authors completely have ignored the idea of suspended load and bed load in debris flow. There are several studies which confirm about their interrelationship for eg.

1) Egashira and Ashida (1992) “Unified view of the mechanics of debris flow and bed-load” Studies in Applied Mechanics.

2) Theule, J. I., et al. "Channel scour and fill by debris flows and bedload transport." Geomorphology 243 (2015): 92-105, etc.

Further some studies related to bedload and suspended load.

1) Van Rijn, Leo C. "Sediment transport, part I: bed load transport." Journal of hydraulic engineering 110.10 (1984): 1431-1456.

2) Joshi, S., & Xu, Y. J. (2017). Bedload and suspended load transport in the 140-km reach downstream of the Mississippi River avulsion to the Atchafalaya River. Water9(9), 716.

3) "Sediment dynamics in the lowermost Mississippi River." Engineering geology 45.1-4 (1996): 457-479.

4) Joshi, S., & Jun, X. Y. (2018). Recent changes in channel morphology of a highly engineered alluvial river–the Lower Mississippi River. Physical Geography39(2), 140-165].

5) Nittrouer et al. (2012) Mitigating land loss in coastal Louisiana by controlled diversion of Mississippi River sand (nature geoscience).

6) Nittrouer and Viparelli 2014 Sand as a stable and sustainable resource for nourishing the Mississippi River Delta (nature geoscience).

7) Allison, Mead A., et al. "A water and sediment budget for the lower Mississippi–Atchafalaya River in flood years 2008–2010: implications for sediment discharge to the oceans and coastal restoration in Louisiana." Journal of Hydrology 432 (2012).

8) Joshi and Xu (2015) Assessment of suspended sand availability under different flow conditions of the Lowermost Mississippi River at Tarbert Landing during 1973-2013 (Water).

I recommend the authors to reference these and other required papers to make a necessary connection between debris flow and sediment transport procedure in rivers and then narrow down to debris flow. This will also help emphasize on the specific use of Bagnold’s equation in this study.

Furthermore, the whole introduction is quite unsystematic, random, and completely lacking interconnections between paragraphs. The rationale of the study is not convincing at all. Also, as a reader I also had a hard time in grasping the connection between title, objectives, and introduction. They have mentioned nothing about sediment concentration earlier which is one of their sub-objective. What type of insights you want to gain (be specific)? Furthermore, it has not been explained about why the authors are focusing on stony-type debris flow? What exactly does stony type debris flow means?

Also, in objective 2 the authors mention about exploring applicability of Bagnold’s theory by removing the hypothesis of uniform sediment concentration? But nowhere before this hypothesis has been explained. Please make introductory comments by briefly explaining the Bagnold’s theory beforehand

Reply:

Thank you for this suggestion.

Introduction has been improved and partially re-written, introducing some of the suggested papers and other new references, in order to better highlight, as suggested, the following aspects:

I)- connections between debris flow and transport processes, bed load and suspended load

As literature indicates [11, 12, 13, 14], different stress regimes can be obtained, depending on the composition of the mixture, with different dominant effect (i.e. either the inertial stress due to the interstitial fluid or the collision stress due to coarser sediments) affecting flow dynamics. Because many interrelated effects occur there are not standard situations and it is very difficult to identify the passage from different mixtures’ behaviors and transport mechanisms. In our opinion, the existence criteria of debris flow can be well described by the classification scheme of Takahashi [14] (see the scheme of Figure 1.11 in Takahashi [14] and reported below for simplicity of exposure). In particular, according to Takahashi’s classification, if coarse particles are not contained in the flow (at the lowest point on the vertical axis of the figure below), the flow is mere water or slurry flow and the flow regime changes from laminar to turbulent or viceversa depending on the Reynolds number. When the grains concentration becomes larger but still less than about 0.02, the flow contains bed load or suspended load depending on the turbulence and viscosity; particle collision stress also appears but it is small. When the grains concentration becomes larger but less than about 0.2, the flow becomes an immature debris flow, the collision stress dominates only in the lower particle mixture layer. With larger values of coarse particles concentration, the flow becomes a dynamic debris flow because under such a concentration the quasi-static stress cannot be dominant. In this case the possible dominant stresses can be either the particles collision stress (i.e. stony-type debris flows), or the turbulent mixing stress (i.e. muddy-type debris flows) and the viscous stress (i.e. the viscous debris flows).

           Figure taken from Takahashi [14]

Thus, the aforementioned concepts have been shortly described in the introduction of the revised manuscript on page 2 lines 31-53.

Some of the suggested references and others new ones have been added

II)- different approaches also taking into account the  bed load with debris flows, and the attempts to treat the bed and suspended loads also in granular flows.

The aforementioned have been briefly introduced in the revised manuscript on page 2 lines 70-76. Some of the suggested references have been also added

III)- characteristics of stony-type debris flow in comparison to other types of debris flows:

Stony debris-flows are generated especially in mountain areas, and in gravel bedded channels originating in the scree slopes located at the base of rock faces (e.g., in the Dolomites, Northern Italy) [44, 45]. They are usually originated by surface runoff determined by intense rainfall events. In the case in which the percentage of finer sediment is low and the percentage of coarse sediment high, the dominant stresses are the particles collision stress, when higher percentages of fines and lower percentage of coarse sediment are contained in the flow the turbulent mixing stress (for muddy-type debris flows) and the viscous stress (for viscous debris flows) dominate.

This has been shortly described in the Introduction of the revised manuscript as suggested on page 2 lines 48-53.

IV)- the limits of Bagnold’s theory applied to debris flows regarding sediment concentration:

As it is clear from the literature [6, 14, 16, 42, 43], the Bagnold’s theory applied to a gravity-driven flow of coarse grains implies that the concentration is constant across the flow depth. In particulart, [see as an example in 6] Bagnold assumed the same velocity scale and the same length scale for both the dispersive pressure and the shear stress; to apply the Bagnold’s constitutive equations to a stony debris flow, these equations  have been integrated over the depth by assuming the concentration C as constant [14].  

As reported in Takahashi [14], it can be considered that “…Bagnold’s (1954) apparatus consisted of double concentric drums whose inner one was stationary and the outer one rotated so as to create the prescribed shearing rates…….. The torque spring and the manometer attached to the inner drum measured the shear stress and pressure, respectively, within the experimental mixture of grains and fluid. The experimental results showed that when the rotating velocity was small (a small shear stress region) pressure and shear stress varied linearly with the change in shearing rate (du/dz), but when the rotating velocity was large (a large shear stress region) both pressure and shear stress varied proportional to the square of (du/dz). In a small stress region (macro-viscous region) grains merely contributed to the increase in the viscosity of the fluid, whereas in a large stress region (inertial region) the effects of the fluid viscosity became negligibly small, and in between the two regions there was a transitional region.” He hypothesized that the pressure and shear stress in the inertial region was produced by the inter-particle collision. Thus, Bagnold’s results were obtained with fixed concentrations and shear rates.

On the other hand, studies by Takahashi [42, 43] suggested that the main characteristics of a stony-type debris flow would be produced by the frequent collisions between coarse particles, and the effect of interstitial fluid would be negligibly small. For this reason, it is possible to integrate the Bagnold’s constitutive equations by assuming C is constant throughout the depth (this gives equation (3) with equation (4) of the revised manuscript).

In the revised manuscript, in order to avoid to enlarge too much the text, the aforementioned has been briefly reported on page 2 lines 55-64 and on page 3 lines 105-108.

The sentence on page 4   lines 159-163 has been rewritten as follows:

According to Takahashi [14, 42, 48], by considering a stony-type debris flow, for which the effect of interstitial fluid would be negligibly small, and assuming an uniform grains concentration and the shear stress proportional to the square of the vertical gradient of the longitudinal velocity u, the integration of the Bagnold’s constitutive equations, under the boundary condition u=0 at z=0 (with u=longitudinal velocity; z= direction orthogonal to the bed)….

V)-the objectives of the present work and the contribution have been more clearly specified, as suggested, on page 3 lines 115-122.

Reviewer

Regarding the rationale, the authors have compiled a list of studies linked to debris flows in one way or other but they fail to connect their idea of enlisting those studies to the objectives of this study. I guess they want to state that velocity distribution analysis in debris flow has not been carried out yet, however, they completely fail to give this take home message. So, can you please restructure the whole introduction accordingly? In addition, I am also not convinced by simply saying lack of studies was the reason behind this study? So what? What meaningful support this study can give to the mechanism of debris flow in rivers? How will the modification help Bagnold’s equation in the bigger picture of sediment transport?

Reply:

Thank you for this observation.

As suggested, the introduction has been re-structured in order to better highlight the specific contribution of the present work and the limits of existing studies regarding the analysis of velocity distribution. See previous point.

In particular, to more clearly explain the problem, in the revised manuscript also the following sentences have been introduced (page 3 lines 105-114):…. “the studies conducted in this field highlight that the velocity distribution, which depends on the variation of the shear rate within the debris body, can be strongly affected by the grains concentration. ……..……..To authors’ knowledge, no systematic research has been conducted to explore the variation of the grains concentration within the debris body and its effect on the velocity profile. This information could be used for interpreting the flow dynamics in numerical models.”

Reviewer

  1. Material and Methods

Reviewer

P3L113 what exactly is stony-type debris flow? Please explain its statistical parameters with references.

Reply:                                            

Thank you for this observation.

The characteristics of stony-type debris flow also as a function of the concentration have been indicated in the Introduction section ( in the revised manuscript on page 2 lines 48-53).

See also previous point III

Reviewer

P3L113 what is the basis for this assumption?

As it is clear from the literature [6, 14, 16, 42, 43] Bagnold’s theory, applied to a gravity-driven flow of coarse grains, implies that the concentration is constant across the flow depth. Bagnold’s results were obtained with fixed concentrations and shear rates [14, 16]

  The studies conducted by Takahashi [42, 43] suggested that the main characteristics of a stony-type debris flow would be produced by the frequent collisions between coarse particles, and the effect of interstitial fluid would be negligibly small. Thus, the Bagnold’s constitutive equations could be applied and integrated by assuming C is constant throughout the depth [14].

In the revised manuscript this has been shortly explained by modifying the sentence on page 4   lines 159-163, as follows:

“According to Takahashi [14, 42, 48], by considering a stony-type debris flow, for which the effect of interstitial fluid would be negligibly small, and assuming an uniform grains concentration and the shear stress proportional to the square of the vertical gradient of the longitudinal velocity u,…..”

See also previous point IV

Reviewer

P3L115 what does ‘z’ mean? Why are the boundary conditions set to be ‘0’ for both ‘u’ and ‘z’?

In the revised manuscript the meaning of u and of z have been specified.

Reviewer

P4L130-132 It looks like the authors are trying to investigate for ‘maximum possible concentration’ but have they clarified this in the objectives? Also, how will the analysis on maximum possible concentration justify the distribution of sediment concentration? Also, how does the velocity distribution fit in here?

Reply:

The objective is not to estimate the ‘maximum possible concentration’ C* but to investigate and to interpret the variation of the grains concentration C within the debris body.  C* could be estimated by using either experimental data appositely collected or physically-based literature data, as it has been more clearly explained in the revised paper.

In the revised manuscript, in order to better clarify the aforementioned objective, the sentence has been re-written as follows (page 5 lines 175-179):

“While the last one could be estimated by using either experimental data appositely collected or physically-based literature data, more investigations should be conducted to evaluate the grains concentration C and its variation within the debris body.”

Reviewer

What is the basis for/reason behind using the specific values for parameters given in section 2.2? For e.g. Bed slope of 15 degree, diameter 3mm, Discharge 3.51/s (3.51?? – is it cubic m?), sediment layer thickness of 0.1, etc.

Reply:

According to previous works [see as an example in 50, 51] the bed slope and the liquid discharge are also parameters affecting the debris flow behavior. These works indicate that mature debris flow (i.e., with the grains dispersed throughout the entire flow depth) can be obtained for bed slope greater than 15%. Based on this, the experimental conditions used in the present work (and the thickness of the sediment layer) have been selected on the basis of literature’s indications [50, 51], in analogy of experimental conditions of previous works with sediments similar to those used in the present work [33, 39] and after preliminary runs conducted to verify the establishment of these conditions.

In the revised manuscript, the following sentence has been introduced (page 5 line 187-189):

“The experimental conditions were selected after preliminary setting runs and in accordance to indications given by previous literature works [33, 39, 50, 51].”

Furthermore, the units of the discharge have been corrected in the revised manuscript.

Reviewer

It will be better if the authors can get a wider frame for Figure 1 including most of the experimental flume (as much as possible at least). Also please describe the settings in the captions including the meaning for the numbers 31, 32, 34, etc.

Reply:

The planview of the experimental apparatus has been added in Figure 1.

The caption has been improved by adding the setting and the meaning of the numbers

Reviewer

P4L150 please explain EH700 and a brief working mechanism behind it.

Reply:

More details of the instrument EH7000 have been introduced (page 6 lines 215-216).

Reviewer

P4L153 how did you define the width of investigating area?

Reply:

Thank you for this suggestion

The width of the investigated area was defined on the basis of the camera characteristics and the rate of the acquisition frequency. Thus, the following sentence has been introduced in the revised manuscript (page 6 lines 218-220):

“The rate of the acquisition frequency was defined, during preliminary runs, as function of the camera’s characteristics and the extension of the covered area.”

Reviewer

In figure 2, please elaborate the caption by explaining the x and y-axes. Also please enlarge figure 2 such that numbers and axes titles are seen more clearly. Also I am not sure if the mechanism behind Figure 2 has been clearly explained anywhere.

Reply:

Thank you for this suggestion

The caption of Figure 2 has been improved and it has been specified that an “electric oscillating sieve” has been used (page 5 lines 208-209).   

Reviewer

After a complete reading up to page 5 of the manuscript I have a slight idea of what the authors are trying to do. However, can they at least restructure the ideas such that the message is clear in the introduction leading up to the objectives and methods should only be a reflection of their experiment? I believe section 2.1 must be restructured by specifying what is meant to do at first.

Reply:

Thank you for this suggestion

Furthermore, as explained in the previous points I, II, III, IV, V,  Introduction section has been improved better clarifying the objectives.

Furthermore, we would like to underline that Subsection 2.1. was introduced simply to briefly summarize the Bagnold’s application to debris flow. In order to meet reviewer’s suggestion this section 2.1 has been improved in order to better highlight the connections with what described in the Introduction.

In particular, in the revised manuscript the following sentence has been introduced (page 4 lines 131-132):

 “Before to proceed further it seems important to briefly summarize here some peculiar aspects of Bagnold’s theory and on its application to debris flow.”

On page 4 lines 156-158 the following sentence has been introduced:

“Bagnold’s results were obtained with fixed concentrations and shear rates [6, 14, 16]; in accordance with Iverson [4], this would imply that the grains concentration should be specified rather than determined by the grains interaction mechanisms.”

Reviewer

Also, the methods for investigating velocity distribution are either missing or hidden somewhere in section 2.2, can you highlight them clearly? This is one of the main objectives of your study.

Reply:

Thank you for this suggestion

The methods for investigating the flow velocity and its distribution have been clearly described on page 6 lines 244-252 of the revised manuscript.

Reviewer

Also, the methods say nothing about what modifications will be carried out in the Bagnold’s equation although this is mentioned even in the title.

Reply:

To meet this suggestion and to better clarify how Bagnold’s number and the equation applied to debris flows have  been modified, as explained in the previous points, in the revised manuscript, more details have been introduced:

- in the Introduction (see also in the previous points I, II, III);

- at the end of sub-section 2.1. (page 4 lines 131-132;  lines 156-158);

- at the beginning of the sub-section 2.2 (page 5 lines 183-187).

- the titles of the sub-sections 3.1. and 3.2 have been slightly modified

  1. Results

Reviewer

P6L181-186 these are more assumptions than actual quantifications. Also what do you mean by static configuration?

Reply:

Thank you for this suggestion

The static configuration occurs at the bed where the frictional stress is so high that no granular motion is possible and the concentration is the maximum one.

 In the revised manuscript (page 7 lines 270-274), the following sentence has been introduced: “…In particular, as Figure 4 clearly shows, as one passes from the free surface to the bed the movement of the grains decreases and their concentration increases; at the bed the frictional stress is so high that no motion is possible (static configuration) and the grains concentration assumes its maximum value (i.e. maximum packing concentration) C*. The thickness of the static bed, zo, was estimated from the analysis both of the recorded images and of the measured velocity profiles.

Reviewer

P6L187-194 these are actually methods according to me, not results

Reply:

According with reviewer’s suggestion, this part of the text has been improved and re-written to better underline the methodology used. In particular lines P6L187-194 of the old manuscript have been moved and re-written at the end of the sub-section 2.2., but the sensitivity analysis, which is in our opinion part of the results obtained in the present work, is maintained in the section 3.1.

Thus, in section 2.2. (page 7 lines 253-260) the methodology used to estimate the grains concentration has been better explained and in section 3.1 (page 8 lines 281-295) the procedure applied to estimate the linear law and the sensitivity analysis have been more clearly explained. In particular the equation (5) was determined after the analysis carried out by assuming the lower value of sub-layer thickness and then the sensitivity analysis was conducted. For this reason a new figure (Figure 5) showing the regression analysis (the other figures have been thus renumbered) has been also introduced in the revised manuscript.

It should also be noted that the symbol of the sub-layer thickness has been changed as “St” in order to avoid confusion with the variable “t” indicating the time which has been introduced in the revised manuscript.

Reviewer

Where is equation (5)?

Reply:

Eq. 5  was indicated in the old manuscript but probably in the transformation in pdf format the number went inside the equation. In the revised manuscript the numbering of equations has been corrected. 

Reviewer

To me section 3.1 mostly presents methods rather than results. I would like to see the actual quantification of grains concentration distribution here rather than the theory and equations behind the distributions.

Results section for showing the statistics which have been revealed after analysis and not the equations. For e.g. a detailed explanation of Figure 5 will be results but the mechanism behind it will certainly highlight methods.

Reply:

See also previous point

According with reviewer’s suggestion, this part of the text has been improved and re-written to better underline the methodology used. In particular (see also previous point), lines P6L187-194 of the old manuscript have been moved and re-written at the end of the sub-section 2.2., but the sensitivity analysis, which is in our opinion is part of the results obtained in the present work, is maintained in the section 3.1.

Thus, in section 2.2. (page 7 lines 253-260) the methodology used to estimate the grains concentration has been better explained and in section 3.1 (page 8 lines 281-295) the procedure applied to estimate the linear law and the sensitivity analysis have been more clearly explained. In particular the equation (5) was determined after the analysis carried out by assuming the lower value of sub-layer thickness and then the sensitivity analysis was conducted. For this reason a new figure (Figure 5) showing the regression analysis (the other figures have been renumbered) has been also introduced in the revised manuscript.

The figure 5 of the old manuscript becomes figure 6 of the revised manuscript. Figure 6  of the revised manuscript has been better explained on the basis of the aforementioned improvements.   

Reviewer

Please elaborate all the notations in figure 5 in the caption (please do this for all figures consistently).

Reply:

Figure 5 of the old manuscript becomes figure 6 of the revised manuscript. The captions of Figure 6 and of other figures have been improved explaining the notations and the axes, as suggested

Reviewer

The same problem of intermixing between results and methods persists throughout the rest of the results section to me. Further problem also exists on lack of or small systematic description on the figures which highlight most of the results.

Also, a discussion section needs to be separated from conclusions for discussing about implications and importance of results that you report in your study, comparing your results with other study results, what they are missing, how your study fills the gap, also in your study what is the importance of the results, what is the reason behind getting those results.

Reply:

Thank you.

In the revised manuscript, in the results section, the figures captions and their description have been improved.

Furthermore, a new section “Discussion” has been added so that the section “Discussion and concluding remarks” has been splitted into the section “Discussion” and the section “Conclusion”, as suggested.

Reviewer 2 Report

Title: Experimental analysis of velocity distribution in a coarse-grained debris flow: a modified Bagnold’s equation

Manuscript ID: water-743941

Overall recommendation: Accept after minor revision

This manuscript deal with the sediment transport on coarse-grained debris flows. Experimental data are used to model a vertical distribution of the grain concentration resulting in a linear relation that depends on the maximum possible concentration and the free surface sediment concentration. This linear expression is used to improve the Bagnold equation. Furthermore, literature data are used to find out a method to obtain the free surface concentration. The work is original, has scientific soundness and it is of interest for the journal readership. I only hace some minor comments before its possible publication:

  1. The numbering of equations has to be revised
  2. Figures with channel photographs has to be edited to improve their quality. I strongly recommend their enlargement. 
  3. When the linear expression for the sediment concentration is presented, the surface concentration is obtained using the averaged value of three samples taken during the experiments. However, later in the manuscript a section is dedicated to the assessment of this surface concentration using literature data. This is perfectly fine for me, but some readers can find this hard to understand. I would explain that with more detail in Section 2.2

Author Response

Reply to reviewers

First of all, Authors wish to thank reviewers for their comments and suggestions which allowed the improvement of the manuscript in its revised form

Original comments by the reviewers are in normal font; reply is in blue and italics. The numbers indicating the references cited in this reply (and not added here) can be found in the revised manuscript 

 Reply to reviewer 2:

Reviewer

This manuscript deal with the sediment transport on coarse-grained debris flows. Experimental data are used to model a vertical distribution of the grain concentration resulting in a linear relation that depends on the maximum possible concentration and the free surface sediment concentration. This linear expression is used to improve the Bagnold equation. Furthermore, literature data are used to find out a method to obtain the free surface concentration. The work is original, has scientific soundness and it is of interest for the journal readership. I only hace some minor comments before its possible publication.

Reply

Thank you very much

1.The numbering of equations has to be revised

Reply:

Thank you for this suggestion.

The numbering of the equations has been corrected, as suggested 

Reviewer

2.Figures with channel photographs has to be edited to improve their quality. I strongly recommend their enlargement. 

Thank you.

Some figures with channel photographs have been enlarged and improved

Reviewer

  1. When the linear expression for the sediment concentration is presented, the surface concentration is obtained using the averaged value of three samples taken during the experiments. However, later in the manuscript a section is dedicated to the assessment of this surface concentration using literature data. This is perfectly fine for me, but some readers can find this hard to understand. I would explain that with more detail in Section 2.2

Reply:

Thank you for this suggestion

In order to clearly distinguish the measured and the estimated free surface concentration the following improvements have been introduced:

- in section 2.2 of the revised paper (page 6 line 224) it has been specified that the value of Cs obtained by the three samples during the passage of the debris body can be considered as a “measured free surface concentration”…and indicated as “Cs,m”.

Thus the sentence on page 6 line 224 has been modified as follows:

“……Cs, was assumed as the mean value of sediment concentration of the three samples and a value of Cs,m =0.305 ….”

-Analogously on page 14 line 523 the symbol has been changed as Cs,m=0.305.

-More details about the importance in the estimation of free surface  grains concentration have been introduced through the text (sections 2.2, 3.2), in the Abstract and in the new “Discussion” section.  

Round 2

Reviewer 1 Report

There are major improvements in the manuscript. I liked many improvements like clarification of rationale and objectives, explanation behind forming experimental values for parameters in methods, addition of discussion section, and elaboration of figure notations. However, to me it still has shortcomings in the technical writing like long complex sentences, unconnected paragraphs, redundant information etc. I can recommend the publication of this manuscript if these technical writing issues are sorted out. Here are few more specific comments as well which I can see to be improved in the manuscript:

The abstract still lacks rationale. 1 or 2 lines of rationale definitely needed

L34: please use a different word than ‘rheological’ properties. Something simpler

L35-39 too long and complex sentence. Can be divided into 2 simpler sentences to convey clearer meaning. In fact, this problem of too complex sentences persists throughout the manuscript.

L42-L47 a sentence of 5 lines is not good for peer-reviewed manuscript.

I guess the results in section 3.1 are based on the experimental setup of the laboratory flume. And section 3.2 is based on section 3.1 and section 3.3 looks like is based on equation 7 and 8. Can these be shown as flow charts in methods. Right now, as a reader I couldn’t separate the results I expected through the methods given.

Section 4 can be part of discussion as well (as a separate sub section)

Discussion seems to be a good addition (however, the language needs to be a bit simpler). Conclusions can be shortened by 1 or 2 points.

Author Response

Reply

First of all, Authors wish to thank reviewer for these second round of suggestions.

Reviewer 1

There are major improvements in the manuscript. I liked many improvements like clarification of rationale and objectives, explanation behind forming experimental values for parameters in methods, addition of discussion section, and elaboration of figure notations. However, to me it still has shortcomings in the technical writing like long complex sentences, unconnected paragraphs, redundant information etc. I can recommend the publication of this manuscript if these technical writing issues are sorted out.

Reply

The English form of the text has been revised by a native English speaking teacher. Where possible, the longer sentences have been shortened, the text has been improved also reducing the redundant information

Here are few more specific comments as well which I can see to be improved in the manuscript:

The abstract still lacks rationale. 1 or 2 lines of rationale definitely needed

Reply

Thank you for this suggestion

A sentence better clarifying the rationale of this study has been introduced at the beginning of the abstract (as consequence the second sentence has been modified in order to maintain the maximum of 200 words required by the journal).

L34: please use a different word than ‘rheological’ properties. Something simpler

Reply

The word “rheological” properties has been changed as “internal stress distribution” 

L35-39 too long and complex sentence. Can be divided into 2 simpler sentences to convey clearer meaning. In fact, this problem of too complex sentences persists throughout the manuscript.

Reply

Thank you for this suggestion

The sentence has been divided into two sentences. 

L42-L47 a sentence of 5 lines is not good for peer-reviewed manuscript.

Reply

Thank you for this suggestion

The sentence has been divided into three sentences

I guess the results in section 3.1 are based on the experimental setup of the laboratory flume. And section 3.2 is based on section 3.1 and section 3.3 looks like is based on equation 7 and 8. Can these be shown as flow charts in methods. Right now, as a reader I couldn’t separate the results I expected through the methods given.

Reply

Thank you for this suggestion

A flow chart has been added (new Figure 14) to explain how to apply the procedure and the application of the proposed expressions

Section 4 can be part of discussion as well (as a separate sub section)

Reply

Thank you for this suggestion

As suggested, section 4. has been included as a sub-section (sub-section 4.1) of the section “Discussion”, which in the revised manuscript includes the aforementioned sub-section 4.1. and the sub-section 4.2. As consequence, the in the revised manuscript the number of the section “Conclusion” has been changed.

Discussion seems to be a good addition (however, the language needs to be a bit simpler). Conclusions can be shortened by 1 or 2 points.

Reply

Thank you for this suggestion

Conclusion has been shortened for one point as suggested

Furthermore, the English form of the text has been revised by a native English speaking teacher.

Reviewer 2 Report

The authors have addressed all my comments 

Author Response

Reply

Reviewer 2

The authors have addressed all my comments 

Reply

Thank you for the positive comment
